# Unleash The Power of Color for Point Cloud Registration

## Abstract

Point cloud registration (PCR) has been an important research subject for many years but remains an open problem, presenting numerous challenges. The stability of existing registration methods is often inadequate, particularly in scenarios with low overlap. This issue primarily arises from the insufficient distinctiveness of extracted point cloud features, leading to ambiguous matches and the proliferation of outliers. To address these bottlenecks in point cloud registration, it is crucial to fully leverage the color information of the point clouds to discern point correspondences effectively. However, excessive control over color may disrupt the spatial structure of the point cloud, making it essential to find a balance between the aggressiveness and stability of color integration. To tackle these challenges, we propose UPC-PCR, which unlocks the potential of color information while maintaining stability. Specifically, we design a Curvature-Color Fusion Module (CCF) to initialize distinctive features. Additionally, to balance color aggressiveness, we enhance the geometric structure by introducing a Centroid Angular (CA) embedding for superpoint structure encoding, which is particularly effective in low-overlap scenes. While CCF and CA ensure the distinctiveness of point features, the aggressive use of color in the feature enhancement process may still introduce errors. Therefore, we develop a robust estimator equipped with Feature-based Compatibility Hypergraph Convolution (FCH) to learn higher-order compatibility of correspondences and effectively filter out outliers. Evaluation across multiple datasets has demonstrated the state-of-the-art performance of UPC-PCR, achieving registration recalls of **98.4%/90.4%** on Color3DMatch/Color3DLoMatch.

## 1 Introduction

Point cloud registration (PCR) is an important yet challenging area in the fields of 3D vision and robotics (Choy et al., 2019; Bai et al., 2020; Huang et al., 2021; Qin et al., 2022; Bai et al., 2021b; Zhang et al., 2023). The objective of PCR is to estimate a rigid transformation that aligns the two input point clouds. Classical PCR algorithms include Iterative Closest Point (ICP) and its variants (Besl & McKay, 1992; Rusinkiewicz & Levoy, 2001; Men et al., 2011; Joung et al., 2009), which minimize the Euclidean distance between corresponding points through iterative refinement. However, when the initial pose is inaccurate, these methods often converge to local optima.

In recent years, there has been rapid development in learning-based methods, particularly those based on feature matching (Ao et al., 2023; Huang et al., 2021; Yang et al., 2022; Yu et al., 2021; 2023b; Qin et al., 2022), leading to significant improvement in registration accuracy. These methods typically utilize neural networks to extract point-wise features and establish correspondences, followed by robust estimators (Bai et al., 2021b; Zhang et al., 2023; Yao et al., 2023; Fischler & Bolles, 1981) to compute the final transformation. To extract prominent local neighborhood information, some methods (Yew & Lee, 2022; Qin et al., 2022; Yu et al., 2023a;b; Chen et al., 2023) utilize downsampling techniques to obtain hierarchical points, from dense points to superpoints. Then, they apply positional embedding (Yang et al., 2022) to superpoints to encode structural information and input them into a transformer to obtain superpoint features.

However, despite improvements in registration performance, it remains insufficient for achieving stable registration in real-world complex scenarios, particularly in challenging situations with low point cloud overlap. To overcome the bottlenecks in PCR, some methods have begun to utilize color

information to assist in the registration process. FCGF (Choy et al., 2019) first explored a learning-based color-assisted PCR method. However, due to the potential disruption of the spatial structure by color information, it did not achieve satisfactory performance. Other methods perform 2D-3D multi-modal learning to eliminate feature ambiguity, such as using image features to enhance point cloud features (Zhang et al., 2022; Yu et al., 2023b; El Banani & Johnson, 2021; Yuan et al., 2023; Wang et al., 2022). However, their multi-modal information interaction is indirect and insufficient, so the potential of color is not fully unlocked. Furthermore, ColorPCR (Mu et al., 2024) achieves a breakthrough in registration performance through multi-stage geometric-color fusion. Since color can disrupt the structure of point cloud, it chooses a relatively conservative approach of multi-step geo-color fusion, which imposes certain limitations on the utilization of color power. To fully Unleash the power of color for PCR, we propose a Curvature-Color Fusion Module (CCF) to provide high-specificity and geometry-stable dense point features.

Due to the similar structure of overlapping regions, efficient positional embedding plays a crucial role in overlap detection (Min et al., 2021; Yang et al., 2022). We adopted this technique to simultaneously enhance geometric features for balancing the effect of color information. After superpoint matching and point matching, we obtain correspondences. Although sufficient correct correspondences are included to estimate rigid transformations, the existing robust estimators are not powerful enough to filter out the errors introduced by color. Therefore, we propose a Feature-based Compatibility Hypergraph Convolution (FCH) to retrieve the high-order compatibility between correspondences, which can stably filter out errors and help generate promising hypotheses.

In summary, we propose a method, named UPC-PCR, which fully unlocks the potential of color to break the bottlenecks in PCR. UPC-PCR enables seamless feature flow in the entire registration process, which results in high registration accuracy, even when the overlap between point clouds is low and the geometric structures are similar. Evaluations on multiple datasets have consistently demonstrated the state-of-the-art performance of UPC-PCR. Specifically, it achieves registration recalls of **98.4%/90.4%** on Color3DMatch/Color3DLoMatch (Mu et al., 2024) datasets. In summary, our main contributions are four-fold:

- We design a Curvature-Color Fusion Module (CCF) to initialize point features. CCF preliminarily extracts structure-color features. It significantly eliminates feature ambiguity and lays the foundation for accurate correspondences.

- We propose a structural-aware Centroid Angular (CA) embedding to encode structural information and enhance the geometric features.

- We design a Feature-based Compatibility Hypergraph Convolution (FCH) as a bridge between the preceding network and transformation estimator, sufficiently detecting the high-order correlation between correspondences and successfully rejecting erroneous correspondences.

- UPC-PCR fully unleash the power of color and leads to a breakthrough in PCR, especially in challenging scenarios, where the overlap between two point clouds is low and the geometric structure is highly similar.

## 2 RELATED WORK

**Learning-based PCR methods.** Compared to earlier ICP algorithm (Besl & McKay, 1992) and its variants (Rusinkiewicz & Levoy, 2001; Men et al., 2011; Joung et al., 2009; Korn et al., 2014), learning-based PCR methods have achieved notable success recently. They (Zeng et al., 2017; Deng et al., 2018b;a; Saleh et al., 2020; Ao et al., 2021; Choy et al., 2019) may employ neural networks to learn local descriptors for 3D correspondence search. Or they (Li & Lee, 2019; Bai et al., 2020; Huang et al., 2021) utilize neural networks to detect keypoints to assist the registration process. CofiNet (Yu et al., 2021) utilizes a coarse-to-fine approach to estimate correspondences, while GeoTransformer (Qin et al., 2022) refines the coarse-to-fine process, further improving registration performance.

**2D-3D multi-modal learning.** 2D-3D multi-modal learning has been employed in various 3D vision tasks, including instance segmentation (Hou et al., 2019) and object detection (Qi et al., 2020), etc. Some methods (El Banani & Johnson, 2021; El Banani et al., 2021; Zhang et al., 2022; Wang et al., 2022; Yuan et al., 2023) have started to use features from 2D images to assist the 3D PCR process. For example, PEAL (Yu et al., 2023b) uses 2D priors to calibrate 3D anchor points in overlapping

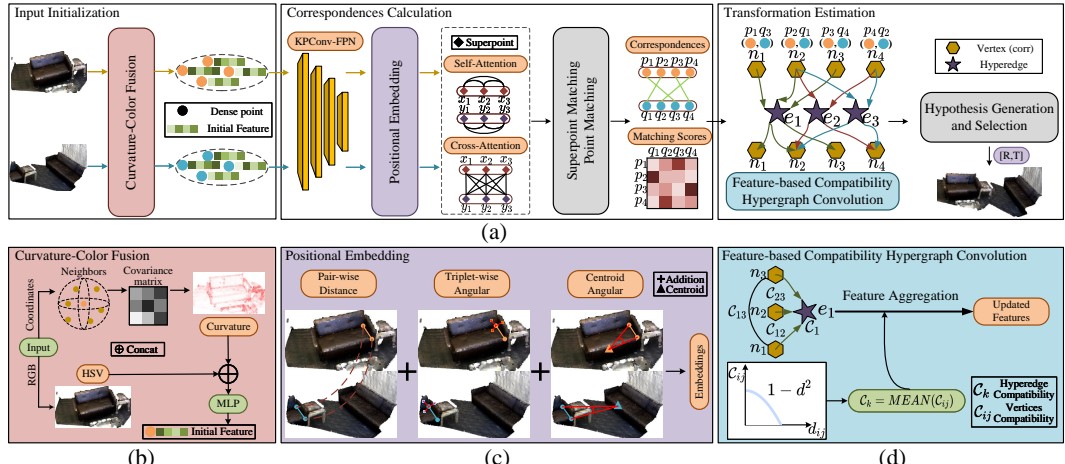

Figure 1: (a) Overview. (b) Curvature-Color Fusion (CCF). (c) Positional embedding. (d) Feature-based Compatibility Hypergraph Convolution (FCH). UPC-PCR takes two color point clouds as input. It first fuses point-wise color and curvature to obtain initial features, and perform hierarchical feature extraction with KPConv-FPN. Then, positional embedding is utilized to encode the structural information of superpoints, followed by a transformer to obtain superpoints' features. Then we follow a a coarse-to-fine matching process to obtain correspondences with matching scores. Ultimately, we utilize corresponding points' features to initialize the FCH for feature aggregate. Based on precise vertex features, hypotheses can be generated and selected to determine the final transformation.

regions, successfully improving registration accuracy. However, their utilization of 2D images is not sufficient, and cannot eliminate feature ambiguity effectively. Recently, ColorPCR (Mu et al., 2024) has proposed a multi-stage geometric-color fusion method for colored point cloud registration, achieving significant breakthrough in PCR performance. Our method follow this way and fully unleash the power of color information, enhancing feature distinctiveness.

**Positional embedding.** Relative positional encoding has been proposed in tasks such as machine translation (Shaw et al., 2018) and has been introduced in PCR. For example, DoPE (Min et al., 2021) utilizes relative positional encoding to iteratively optimize a joint-origin. GeoTransformer (Qin et al., 2022) uses distance and angle encoding to represent the positional information between any two points. Similarly, OIF-PCR (Yang et al., 2022) performs an iterative process to optimize reference points for positional encoding.

**Robust Transformation Estimator.** After obtaining correspondences, a robust estimator is needed for transformation estimation. Traditional methods such as RANSAC (Fischler & Bolles, 1981) perform global random exploration. Recently, MAC (Zhang et al., 2023) proposes to loose the maximum clique constraint and retrieve maximal cliques, and FastMAC (Zhang et al., 2024) accelerates its runtime. Also, some deep robust estimators (Bai et al., 2021b; Yao et al., 2023) have been developed. They use neural networks to reject outliers. However, all these methods are disconnected from the preceding network and perform transformation estimation separately. Instead, UPC-PCR proposes to bridge the two stages with features by leveraging FCH. It retrieves high-order consistency in correspondences, constructing a comprehensive registration network.

## 3 METHOD

### 3.1 PROBLEM STATEMENT

Given source colored point cloud $\mathcal{P} = \{\mathbf{p_i} \in \mathbb{R}^3 | i = 1, ..., N\}$ with $\mathcal{P}_c = \{\mathbf{p_{ci}} \in [0,1]^3 | i = 1, ..., N\}$ and target colored point cloud $\mathcal{Q} = \{\mathbf{q_i} \in \mathbb{R}^3 | i = 1, ..., M\}$ with $\mathcal{Q}_c = \{\mathbf{q_{ci}} \in [0,1]^3 | i = 1, ..., M\}$, the objective of point cloud registration (PCR) is to estimate a rigid transformation $T = \{R, t\}$, where $R \in SO(3)$ and $t \in \mathbb{R}^3$. By applying the transformation on $\mathcal{P}$, the two point clouds can be aligned. The process of estimating the optimal transformation can be defined as solving

the following optimization problem:

$$\min_{\mathbf{R}, \mathbf{t}} \sum_{(\bar{\mathbf{p}}_i, \bar{\mathbf{q}}_i) \in \bar{\mathcal{C}}} \|\mathbf{R} \cdot \bar{\mathbf{p}}_i + \mathbf{t} - \bar{\mathbf{q}}_i\|_2^2, \tag{1}$$

where $\bar{\mathcal{C}}$ is the ground-truth correspondence set between $\mathcal{P}$ and $\mathcal{Q}$.

## 3.2 PIPELINE

For ease of description, we provide the following symbol explanations. Taking the source colored point cloud as an example, We perform a down-sample process on dense points $\mathcal{P} \in \mathbb{R}^{|\mathcal{P}| \times 3}$ to get the multi-level points, from $\tilde{\mathcal{P}} \in \mathbb{R}^{|\tilde{\mathcal{P}}| \times 3}$ to the superpoints (patches) $\hat{\mathcal{P}} \in \mathbb{R}^{|\hat{\mathcal{P}}| \times 3}$, where $|\hat{\mathcal{P}}| < |\tilde{\mathcal{P}}| < |\mathcal{P}|$. The representation for the target colored point cloud $\mathcal{Q}$ is similar.

Fig. 1 illustrates the pipeline of UPC-PCR. We follow prior works (Sun et al., 2021; 2019; Yu et al., 2021; Qin et al., 2022) to perform registration with a coarse-to-fine process (*i.e.* from patch correspondences to point correspondences). We first calculate the curvature of raw points generated from depth images, and get the point-wise color from their corresponding RGB images. Then we utilize the Curvature-Color Fusion Model (CCF) to initialize dense points features (Sec. 3.3), followed by the backbone KPConv-FPN (Thomas et al., 2019; Lin et al., 2017) for hierarchical feature extraction. Next, we apply positional embedding (Sec. 3.4) on superpoints and feed them to a Transformer (Qin et al., 2022) to obtain superpoint features $\hat{\mathcal{F}}_{\mathcal{P}}$ and $\hat{\mathcal{F}}_{\mathcal{Q}}$. Based on these features, patch correspondences can be estimated (Qin et al., 2022). Then an optimal transport layer (Sarlin et al., 2020) is used to obtain the point correspondences from the patch correspondences. To sufficiently retrieve high-order compatibility between correspondences and filter out outliers, we utilize the point-pair features from $\tilde{\mathcal{F}}_{\mathcal{P}}$ and $\tilde{\mathcal{F}}_{\mathcal{Q}}$ to initialize a correspondence hypergraph. Then the Feature-based Compatibility Hypergraph Convolution (Sec. 3.5) can efficiently aggregate vertex features for promising initial hypotheses generation. Finally, we follow Hunter (Yao et al., 2023) to perform local exploration on the initial hypothesis to alternatively generate transformations, followed by final transformation selection.

## 3.3 CURVATURE-COLOR FUSION MODEL

Feature initialization has been proven critical in colored point cloud registration (Zhang et al., 2022). However, inappropriate multi-modal fusion methods cannot effectively eliminate feature ambiguity. Therefore, we propose the CCF model to explicitly initialize point-wise features. We follow (Rusu & Cousins, 2011) to compute curvature. For each point $\mathbf{p_i}$ in the colored point cloud $\mathcal{P}$, with its $r$-radius neighborhood denoted as $\mathcal{B}_r^3 = \{\mathbf{x} \in \mathbb{R}^3 \mid \|x\| \leq r\}$, we compute its covariance matrix $C \in \mathbb{R}^{3 \times 3}$:

$$C = \frac{1}{N-1} \sum_{i=1}^{N} (\mathbf{p_i} - \bar{\mathbf{p}})(\mathbf{p_i} - \bar{\mathbf{p}})^T, \tag{2}$$

where $N$ is the number of points in the neighborhood and the division by $N - 1$ in the formula aims to obtain an unbiased estimate. $\bar{\mathbf{p}}$ is the mean of the points in the neighborhood, i.e., $\bar{\mathbf{p}} = \frac{1}{N} \sum_{i=1}^{N} \mathbf{p_i}$. Then we calculate the eigenvalues $\lambda_1 < \lambda_2 < \lambda_3$ of the covariance matrix $C$. The final curvature can be computed by $k = \frac{\lambda_1}{\lambda_1 + \lambda_2 + \lambda_3}$.

For points color, since the raw 3DMatch dataset (Zeng et al., 2017) is composed of RGB-D images, color can be simply acquired from RGB images. ColorPCR (Mu et al., 2024) preprocessed the 3DMatch in this way and achieved Color3DMatch, where every point contains an RGB value. We utilize Color3DMatch and follow ColorPCR to convert RGB into the HSV color system. Finally, we simply concatenate the curvature and the color, and employ a multi-layer perception to fuse them and adjust the feature dimension to match the following network modules.

## 3.4 POSITIONAL EMBEDDING

Since the local features of point clouds may not be prominent, relying solely on KPConv-based feature extraction is insufficient to eliminate feature ambiguity. By introducing positional embedding, spatial consistency can be preserved and therefore improve the registration performance (Min et al.,

2021; Yang et al., 2022; Qin et al., 2022). Given two pairs of corresponding points under ground-truth, denoted as $(\mathbf{p_x}, \mathbf{q_a})$ and $(\mathbf{p_y}, \mathbf{q_b})$, where $\mathbf{p_x}, \mathbf{p_y} \in \mathcal{P}$ and $\mathbf{q_a}, \mathbf{q_b} \in \mathcal{Q}$, we adopt three types of embedding to encode the geometric structure of the superpoints based on structural properties they have. The embeddings are named Pair-wise Distance (PD) embedding (Qin et al., 2022), Triplet-wise Angular (TA) embedding (Qin et al., 2022)), and Centroid Angular (CA) embedding. And we use the sum of them as the final embedding.

(1) Pair-wise Distance (PD) Embedding $r^D$ (Qin et al., 2022). It is based on the structural property that for correspondence pairs $(\mathbf{p_x}, \mathbf{q_a})$ and $(\mathbf{p_y}, \mathbf{q_b})$, we have $\|\mathbf{p_x} - \mathbf{p_y}\| = \|\mathbf{q_a} - \mathbf{q_b}\|$. Denoting the three-dimensional Euclidean distance between $(\mathbf{p_x}, \mathbf{q_a})$ as $d = \|\mathbf{p_x} - \mathbf{q_a}\|$, we have $r^D = F(\frac{d}{\sigma_d})\mathbf{W^D}$, where $F$ is a sinusoidal function (Vaswani et al., 2017) and $\sigma_d$ is a hyperparameter used to tune the sensitivity on distance variations. $\mathbf{W^D} \in \mathbb{R}^{d_t \times d_t}$ is the corresponding projection matrix, where $d_t$ is the output dimension of $F$. Notably, ColorPCR (Mu et al., 2024) introduce color information by multiplying $d$ with the hue difference between $\mathbf{p_x}$ and $\mathbf{q_a}$ (named hue-PD) and achieved an improvement. For more details, refer to our ablation experiments in Sec. 4.3.

(2) Triplet-wise Angular (TA) Embedding $r^A$ (Qin et al., 2022). For any $\mathbf{x_i}$ among the $k_a$ nearest neighbors of $\mathbf{p_x}$, forming an angle $\angle ixy$, the largest $\angle ixy = \angle jab$, where $j$ is the index of the largest angle in the neighborhood of $q_a$. Based on this property, we denote $\angle ixy$ as $a_i$, and its TA embedding can be computed as $r^A = \max_{\mathbf{x_i}}[F(\frac{a_i}{\sigma_a})\mathbf{W^A}]$, where $\sigma_a$ is a hyperparameter used to tune the sensitivity on angular variations and $\mathbf{W^A} \in \mathbb{R}^{d_t \times d_t}$ is the corresponding projection matrix.

(3) Centroid Angular (CA) Embedding $r^{CA}$. We specially design CA to assist in the extremely low overlap registration cases, as shown in Figure 2. For any two points $\mathbf{p}$, $\mathbf{q}$ in the overlapping regions, they can form centroid angles $\angle pc_1q$ and $\angle pc_2q$ with centroids $\mathbf{c_1}$ and $\mathbf{c_2}$. Denoting $\alpha = \max_{\mathbf{pq}} \angle pc_1q$ and $\beta = \max_{\mathbf{pq}} \angle pc_2q$, we claim that any centroid angle in the overlapping region is less than $\theta = max\{\alpha, \beta\}$. When $\theta$ is sufficiently small, the CA embeddings of the overlapping region are approximately equal. To validate this hypothesis, we conduct detailed ablation experiments (Sec. 4.3) and test UPC-PCR under extremely low overlap conditions (Figure 3). The experimental results confirm our assumption. Denoting any centroid angle as $C_a$, then its CA embedding can be computed as $r^{CA} = F(\frac{C_a}{\sigma_{ca}})\mathbf{W^{CA}}$, where $\sigma_{ca}$ is a hyperparameter used to tune the sensitivity and $\mathbf{W^{CA}} \in \mathbb{R}^{d_t \times d_t}$ is the corresponding projection matrix.

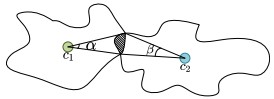

Figure 2: Centroid Angular embedding for extremely low overlap scenarios.

### 3.5 Feature-based Compatibility Hypergraph Convolution

Although CCF and CA can effectively detect many correct correspondences in most cases, they inevitably introduce additional errors. Therefore, we design a hypergraph convolution to learn higher-order compatibility among correspondences for outlier rejection. The hypergraph we utilize here can be denoted as $G = (V, E)$, where $V$ represents the set of vertices, and $E$ represents the set of hyperedges. In $V$, each vertex $v_i = (\mathbf{p_i}, \mathbf{q_i})$ is a correspondence (*i.e.* a pair of corresponding points) predicted in the previous module, with the correspondence matching scores denoted as $M = \{m_i | m_i \in [0, 1], 1 \le i \le N\}$, where $N$ is the number of correspondences.

A hypergraph can be represented as an incidence matrix $H$. If $H(i, j) = 1$, it indicates that the hyperedge $e_j$ connects vertex $v_i$; otherwise, $H(i, j) = 0$. The degree of vertex $v_i$ and hyperedge $e_j$ are defined as $D(v_i) = \Sigma_{j=1}^{|E|} H(i, j)$ and $D(e_j) = \Sigma_{i=1}^{|V|} H(i, j)$. Compared to graphs, Hypergraphs allow hyperedges to connect more than one vertices, which satisfy compatibility. By using the hypergraph convolution for vertex feature aggregation, the high-specificity features from the preceding network can be fused with spatial consistency, which can facilitate more accurate initial hypotheses.

We still use the symbols introduced in Sec. 3.4 and take the correspondences $v_i = (\mathbf{p_x}, \mathbf{q_a})$ and $v_j = (\mathbf{p_y}, \mathbf{q_b})$ as examples. The compatibility between them can be computed by $\mathcal{C}_{ij} = 1 - (\frac{d}{\sigma_c})^2$, where $d = |\|\mathbf{p_x} - \mathbf{p_y}\| - \|\mathbf{q_a} - \mathbf{q_b}\||$ and $\sigma_c$ is a hyperparameter for tuning the sensitivity of the compatibility. Then, we introduce the compatibility of hyperedges. For hyperedge $e_k$, its

compatibility can be computed as:

$$\mathcal{C}_k = \Sigma_{i=1}^{|V|} \Sigma_{j=i}^{|V|} \frac{H(i,k) \cdot H(j,k) \cdot \mathcal{C}_{ij}}{D(e_k)}. \tag{3}$$

Each vertex will induce a hyperedge, and for the hyperedge induced by vertex $v_i$, it will connect to the top $k_c$ vertices with the highest compatibility to $v_i$. In this way, we complete the construction of the compatibility hypergraph. Next, we introduce a hypergraph convolution similar to (Yao et al., 2023) to perform vertex feature aggregation. For any vertex (correspondence) $v_i = (\mathbf{p_x}, \mathbf{q_a})$, features $\mathbf{f_x} \in \tilde{\mathcal{F}}_{\mathcal{P}}$ and $\mathbf{f_a} \in \tilde{\mathcal{F}}_{\mathcal{Q}}$ from backbone are used for hypergraph initialization. Specifically, the feature of this vertex $v_i$ can be initialized by concatenating $\mathbf{f_x}$ and $\mathbf{f_a}$.

The feature aggregation process can be represented as follows. For any hypergraph convolution layer $l$, the feature of vertex $v_i$ at layer $l$ is denoted as $\mathbf{F}_i^l \in \mathbb{R}^{d_l}$, it can be updated as

$$\hat{\mathbf{F}}_i^l = \Sigma_{j=1}^{|V|} \alpha_{ij} m_j \mathbf{F}_j^l \mathbf{W}^l, \tag{4}$$

where $\mathbf{W}^l \in \mathbb{R}^{d_l} \times \mathbb{R}^{d_{l+1}}$ is corresponding projection matrix, $m_j$ is the matching score of $v_j$ and $\alpha_{ij}$ measuring the correlation between $v_i$ and $v_j$ can be computed as:

$$\alpha_{ij} = \frac{1}{\sqrt{D(v_i)D(v_j)}} \Sigma_{k=1}^{|E|} \frac{H(i,k) \cdot H(j,k) \cdot \mathcal{C}_k}{D(e_k)}. \tag{5}$$

Through this aggregation process, spatial consistency propagates from hyperedges to vertices. Then we compute the inlier confidence $\mathbf{S}^l$ of layer $l$ by successively applying an MLP and sigmoid function on $\hat{\mathbf{F}}^l$. To normalize the features, we compute the weighted mean $\mu^l = \Sigma_{i=1}^{|V|} \mathbf{S}_i^l \hat{\mathbf{F}}_i^l / \Sigma_{i=1}^{|V|} \mathbf{S}_i^l$ and standard deviation $\sigma^l = \sqrt{\frac{1}{|V|} \Sigma_{i=1}^{|V|} (\hat{\mathbf{F}}_i^l - \mu^l)^2}$ of them. The normalized features of layer $l$ is computed as $\overline{\mathbf{F}}^l = (\hat{\mathbf{F}}^l - \mu^l)/\sigma^l$. Finally, The feature of next layer $\mathbf{F}^{l+1}$ can be calculated by applying a ReLU function on $\overline{\mathbf{F}}^l$. In the rest steps, we follow (Yao et al., 2023) to estimate the transformation, including Initial Hypotheses Generation based on Non-Maximum Suppression (Lowe, 2004), Preference-Based Local Exploration, and Distance-Angle Based Hypothesis Selection.

## 3.6 LOSS FUNCTIONS

To train our model, the loss functions we use include overlap-aware circle loss ($\mathcal{L}_{oc}$) and point matching loss ($\mathcal{L}_p$) from GeoTransformer (Qin et al., 2022), spectral matching loss ($\mathcal{L}_{sm}$) proposed in PointDSC (Bai et al., 2021b), and classification loss ($\mathcal{L}_{class}$) introduced in Hunter (Yao et al., 2023). The final loss can be obtained through a weighted sum of the four loss components. Notably, we find that using a two-stage approach can yield a slight performance improvement compared to end-to-end training, as detailed in Sec. 4.3.

## 4 EXPERIMENTS

We evaluate the performance of UPC-PCR on the colored point cloud datasets Color3DMatch/Color3DLoMatch (Zeng et al., 2017; Huang et al., 2021; Mu et al., 2024), as well as the ScanNet (Dai et al., 2017a). In Color3DMatch, the overlap between colored point cloud pairs exceeds 30%, while in COlor3DLoMatch, it ranges from 10% to 30%. ScanNet is a large-scale dataset comprising 1513 scenes, and each scene contains RGB-D images and ground-truth camera poses.

## 4.1 COLOR3DMATCH AND COLOR3DLOMATCH

**Metrics.** Similar to prior works (Huang et al., 2021; Qin et al., 2022; Yu et al., 2023b; Mu et al., 2024), we evaluate five metrics including the Inlier Ratio (IR), which measures the proportion of correctly estimated point correspondences during registration (with residuals less than a distance threshold of 0.1m under the ground truth transformation); Feature Matching Recall (FMR), which assesses the proportion of point cloud pairs with IR greater than a threshold (5%); and the most crucial metric, Registration Recall (RR), defined as the proportion of point cloud pairs correctly

Table 1: Results on Color3DMatch and Color3DLoMatch. #Samples in the table represents the number of the selected correspondences. The "color source" column in the table indicates the origin of the color. "concatenate color" indicates that the method does not use color and we introduce color to them in a simple concatenation (refer to appendix) manner to facilitate fair comparison; "RGB-D images" signifies that the method employs images to assist in point cloud registration; "colored PC" indicates that color comes from colored point cloud. The best scores are in **bold**.

| #Samples | color source | Color3DMatch | | | | | Color3DLoMatch | | | | |
|---|---|---|---|---|---|---|---|---|---|---|---|
| | | 5000 | 2500 | 1000 | 500 | 250 | 5000 | 2500 | 1000 | 500 | 250 |
| Feature Matching Recall (%)↑ | | | | | | | | | | | |
| Predator (Huang et al., 2021) | condatenate color | 97.5 | 97.8 | 97.4 | 97.2 | 96.6 | 77.8 | 78.5 | 79.8 | 79.1 | 78.7 |
| CoFiNet (Yu et al., 2021) | condatenate color | 99.1 | 98.9 | 99.1 | 99.0 | 99.2 | 85.8 | 86.2 | 86.1 | 86.4 | 86.5 |
| GeoTransformer (Qin et al., 2022) | condatenate color | 98.7 | 98.6 | 98.9 | 98.7 | 98.9 | 90.2 | 90.3 | 90.1 | 90.2 | 90.2 |
| PCR-CG (Zhang et al., 2022) | RGB-D images | 97.4 | 97.5 | 97.7 | 97.3 | 97.6 | 80.4 | 82.2 | 82.6 | 83.2 | 82.8 |
| PEAL (Yu et al., 2023b) | RGB-D images | 99.0 | 99.0 | 99.1 | 99.1 | 98.8 | 91.7 | 92.4 | 92.5 | 92.9 | 92.7 |
| ColorPCR (Mu et al., 2024) | colored PC | 99.5 | 99.5 | 99.5 | 99.5 | 99.5 | **96.5** | 96.5 | **97.0** | **97.0** | **96.7** |
| UPC-PCR (ours) | colored PC | **99.8** | **99.8** | **99.8** | **99.8** | **99.8** | 96.3 | **96.5** | 96.5 | 96.8 | **96.7** |
| Inlier Ratio (%)↑ | | | | | | | | | | | |
| Predator (Huang et al., 2021) | condatenate color | 52.4 | 53.4 | 52.8 | 50.8 | 46.5 | 21.6 | 23.6 | 24.8 | 24.4 | 23.1 |
| CoFiNet (Yu et al., 2021) | condatenate color | 54.1 | 55.5 | 56.2 | 56.5 | 56.3 | 27.8 | 29.6 | 30.6 | 30.9 | 31.1 |
| GeoTransformer (Qin et al., 2022) | condatenate color | 75.8 | 76.3 | 77.2 | 84.2 | 87.4 | 44.5 | 46.7 | 47.9 | 53.4 | 58.8 |
| PEAL (Yu et al., 2023b) | RGB-D images | 72.4 | 79.1 | 84.1 | 86.1 | 87.3 | 45.0 | 50.9 | 57.4 | 60.3 | 62.2 |
| ColorPCR (Mu et al., 2024) | colored PC | **75.0** | **80.5** | **84.7** | **86.5** | **87.8** | **51.2** | **56.6** | **63.1** | **66.0** | **68.0** |
| UPC-PCR (ours) | colored PC | 71.4 | 77.6 | 82.4 | 84.5 | 86.0 | 47.2 | 53.1 | 59.8 | 62.8 | 65.1 |
| Registration Recall (%)↑ | | | | | | | | | | | |
| Predator (Huang et al., 2021) | condatenate color | 90.7 | 90.6 | 89.6 | 90.8 | 84.8 | 60.1 | 61.8 | 62.6 | 62.8 | 58.2 |
| CoFiNet (Yu et al., 2021) | condatenate color | 90.8 | 91.5 | 91.3 | 91.1 | 90.5 | 65.2 | 66.3 | 65.5 | 66.0 | 65.5 |
| GeoTransformer (Qin et al., 2022) | condatenate color | 94.3 | 93.7 | 93.7 | 93.9 | 93.4 | 81.7 | 81.2 | 80.8 | 80.4 | 80.1 |
| PCR-CG (Zhang et al., 2022) | RGB-D images | 89.4 | 90.7 | 90.0 | 88.7 | 86.8 | 66.3 | 67.2 | 69.0 | 68.5 | 65.0 |
| PEAL (Yu et al., 2023b) | RGB-D images | 94.6 | 93.7 | 93.7 | 93.9 | 93.4 | 81.7 | 81.2 | 80.8 | 80.4 | 80.1 |
| ColorPCR (Mu et al., 2024) | colored PC | 96.7 | 96.5 | 97.0 | 96.4 | 96.5 | 88.9 | 88.5 | 88.1 | 86.5 | 85.0 |
| UPC-PCR (ours) | colored PC | **98.3** | **98.4** | **98.0** | **97.6** | **97.0** | **90.4** | **90.1** | **89.5** | **87.0** | **85.6** |

registered (with a transformation error RMSE less than a threshold of 0.2m). We also evaluate the Relative Rotation Error (RRE) and Relative Translation Error (RTE), which measure the quality of the estimated transformation.

**Comparison with recent methods.**   We compare UPC-PCR with recent state-of-the-art methods, including, Predator (Huang et al., 2021), CoFiNet (Yu et al., 2021), GeoTransformer (Qin et al., 2022), PCR-CG (Zhang et al., 2022) PEAL (Yu et al., 2023b) and ColorPCR (Mu et al., 2024). The comparison results are shown in Table 1. We report the FMR, IR, and RR with samples of 250, 500, 1000, 2500, and 5000. The FMR metric reflects the specificity of point-wise features, as only correspondences with close distances in the feature latent space are correctly identified as inliers. Therefore, a higher FMR value indicates that the network extracts features more accurately and identifies a greater number of inliers in most cases. Our UPC-PCR achieves the FMR of 99.8%/96.8% on Color3DMatch/Color3DLoMatch, demonstrating the significant effect of the proposed curvature-color fusion model. For IR, because of the aggressive introduction of color, which may disrupt the spatial structure of point cloud, the IR of our UPC-PCR is lower compared to methods like PEAL (Yu et al., 2023b) and ColorPCR (Mu et al., 2024). However, the inlier ratio is not directly correlated with the final registration results (as described in OIF-PCR (Yang et al., 2022)); a relatively high inlier ratio is sufficient to estimate an accurate transformation from the correspondences. By leveraging the powerful high-order correlation learning ability of FCH, UPC-PCR can effectively reject outliers, thereby achieving more accurate registration with higher registration recall (RR). We also investigated the factors influencing IR, as detailed in Sec. 4.3. Regarding RR, the most important metric, our UPC-PCR achieves the best performance on both Color3DMatch and Color3DLoMatch, with rates of 98.4% and 90.4%, respectively. UPC-PCR surpasses previous methods in most samples.

**Registration with challenging overlaps.**   In Table 1, we compare UPC-PCR with recent registration methods, with results on Color3DLoMatch significantly surpassing previous methods. To further validate the robustness of UPC-PCR in cases with extremely low overlap, we divide Color3DLoMatch into four overlap intervals for testing. We compared the results with, CoFiNet (Yu et al., 2021), GeoTransformer (Qin et al., 2022) and ColorPCR (Mu et al., 2024) using RANSAC with 5000 samples. The experimental results are shown in Figure 3. The FMR plot demonstrates that even with

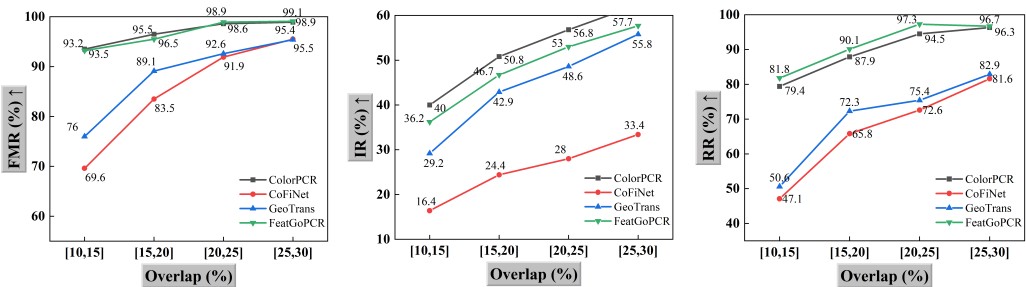

Figure 3: Experimental results of UPC-PCR under different low overlaps.

Table 2: Results on ScanNet. In the table, methods Predator, CofiNet and GeoTransformer do not utilize color originally. We modified them by incorporating color via concatenation (refer to appendix) and retrained them on Color3DMatch. The "color source" column in the table has the same meaning as in Table 1.

| Method | color info | Rotation (deg) | | | | | Translation (cm) | | | | |
| | | Accuracy (%)↑ | | | Error (deg)↓ | | Accuracy (%)↑ | | | Error (cm)↓ | |
| | | 5 | 10 | 45 | Mean | Med. | 5 | 10 | 25 | Mean | Med. |
|---|---|---|---|---|---|---|---|---|---|---|---|
| UR&R (El Banani et al., 2021) (Supervised) | RGB-D images | 92.3 | 95.3 | 98.2 | 3.8 | **0.8** | 77.6 | 89.4 | 95.5 | 7.8 | 2.3 |
| Predator (Huang et al., 2021) | condatenate color | 94.9 | 98.2 | 99.1 | 3.2 | 1.5 | 65.3 | 90.1 | 97.3 | 7.1 | 3.9 |
| CofiNet (Yu et al., 2021) | condatenate color | 95.2 | 98.4 | 99.3 | 2.8 | 1.4 | 68.0 | 91.5 | 98.0 | 6.4 | 3.7 |
| GeoTransformer (Qin et al., 2022) | condatenate color | 96.6 | 98.4 | 99.0 | 2.7 | 0.9 | 81.0 | 93.6 | 97.8 | 6.2 | 2.4 |
| ColorPCR (Mu et al., 2024) | colored PCR | **97.4** | 98.7 | 99.1 | 1.9 | 0.9 | **83.1** | **94.9** | **98.1** | **4.8** | 2.3 |
| UPC-PCR (ours) | colored PCR | **97.4** | **98.8** | **99.5** | **1.8** | **0.8** | **83.1** | 94.4 | **98.1** | **4.8** | **2.2** |

an overlap as low as 10%-15%, UPC-PCR can effectively extract hierarchical point-wise features, achieving a 93.2% FMR. When the overlap exceeds 20%, UPC-PCR achieves approximately 99% FMR, reflecting its powerful ability to eliminate feature ambiguity, and it is hardly affected by the overlap. From the IR plot, we can observe that under extremely low overlaps, UPC-PCR still achieves a relatively high inlier ratio, only slightly lower than ColorPCR. In terms of the most crucial metric, Registration Recall (RR), although the other three methods achieve high accuracy in registration cases with higher overlaps, when the overlap is low, the accuracy of the methods significantly decreases, particularly for CofiNet and GeoTransformer. In comparison, ColorPCR performs relatively well, and our method utilizes color more effectively, consistently outperforming its accuracy. UPC-PCR demonstrates stable performance and achieves an RR of 81.8%. This experimental result corroborates the inference in Sec. 3.4 that CCF and CA can effectively assist in identifying overlapping regions under extremely low overlap conditions.

## 4.2 SCANNET

To test the generalization performance of UPC-PCR, we used the model trained on Color3DMatch and directly evaluated it on ScanNet (Dai et al., 2017a). All comparison methods were trained on Color3DMatch and tested on ScanNet. To ensure a fair comparison, we introduced color to the methods that do not originally include it (Predator, CofiNet, GeoTransformer), using a simple concatenation approach, just the same as in Table 6. We use two evaluation metrics, rotation error and translation error, reporting the mean and median values of the errors. Additionally, we showcase the method's accuracy under three different thresholds. We compare our method with recent point cloud registration methods UR&R (El Banani et al., 2021), Predator (Huang et al., 2021), CofiNet (Yu et al., 2021), GeoTransformer (Qin et al., 2022) and ColorPCR (Mu et al., 2024). The experimental results are shown in Table 2.

## 4.3 ABLATION

As mentioned in Sec. 1, the main contributions of UPC-PCR are the Curvature-Color Fusion Model (CCF), Centroid Angular (CA) Embedding, and Feature-based Compatibility Hypergraph Convolution (FCH), which collectively construct a seamless feature flow to achieve robust point cloud registration. To explore the roles of these components in feature propagation and transformation estimation, we conducted ablation experiments on them respectively.

Table 3: Ablation on CCF.

| Experiment | CCF | | 3DMatch | | | | 3DLoMatch | | | |
|---|---|---|---|---|---|---|---|---|---|---|
| | curvature | color | PIR (%)↑ | FMR (%)↑ | IR (%)↑ | RR (%)↑ | PIR (%)↑ | FMR (%)↑ | IR (%)↑ | RR (%)↑ |
| (a) | ✓ | ✓ | 84.8 | 99.8 | 69.7 | 98.3 | 57.6 | 96.3 | 46.9 | 90.4 |
| (b) | | ✓ | 88.5 | 99.8 | 73.6 | 98.0 | 61.9 | 95.7 | 51.2 | 89.1 |
| (c) | ✓ | | 86.4 | 98.2 | 70.6 | 93.1 | 54.2 | 87.6 | 42.9 | 76.2 |
| (d) | | | 86.2 | 98.6 | 70.7 | 93.9 | 52.6 | 87.6 | 41.9 | 74.6 |

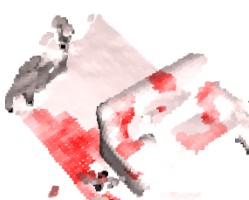 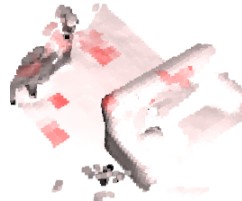 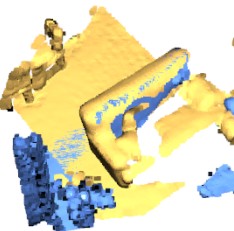 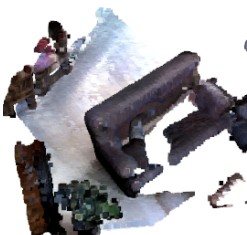

(a) with CCF          (b) w/o CCF          (c) ground truth          (c) ground truth

Figure 4: Visualization of feature matching. Given the points in the overlapping region of the source colored point cloud, figures (a) and (b) visualize the heatmap of the target colored point cloud based on feature similarity. Figures (c) and (d) illustrate the overlapping region of the ground truth, where the blue points represent the source and the yellow points represent the target.

**Ablation on Curvature-Color Fusion model.**    UPC-PCR can extract highly specific features. This heavily relies on the ambiguity elimination in the feature initialization module, CCF. We visualized the feature extraction results of CCF using heatmaps, as shown in Figure 4. We used CCF to extract features from two colored point clouds and selected a set of points $S$ from the overlapping region of the source colored point cloud, determining the maximum similarity of each point in the target colored point cloud to the points in $S$. Points with higher feature similarity were assigned a deeper red color, creating a feature heatmap for the target colored point cloud. As shown in (a) and (b), when using CCF, many points in the overlapping region exhibit high similarity, while others have low similarity. This confirms that the features extracted by CCF have high specificity, and the algorithm assigns higher feature similarity to points in the overlapping region. However, the figure also reflects that CCF can result in some non-overlapping points being assigned high feature similarity. This is due to these points having similar colors. This finding aligns with the observed decrease in the inlier ratio in our experimental results. This is why we designed FCH to search for higher-order associations among corresponding points, filtering out incorrect correspondences for accurate registration.

Furthermore, we conducted ablation experiments on CCF under the same conditions of Triplet-wise Angular embedding, Pair-wise Hue-Distance Embedding, Centroid Angular embedding, and our FCH-based transformation estimator without sampling the correspondences. The experimental results are shown in Table 3. When (d) does not use the CCF module, there is no incorporation of feature initialization information, resulting in lower feature specificity and lower RR values. In experiments (b) and (c), the introduction of feature initialization enhances point-wise features, leading to significant improvements in various metrics of the network. However, in experiment (a) where the CCF module is employed, while FMR and RR are further improved, it leads to a decrease in Patch Inlier Ratio (PIR) and IR. As described in Sec. 3.4, the introduction of aggressive and unstable information may lead to a decrease in PIR and IR, but the overall registration accuracy is not affected. Consider the following facts: (1) The introduction of curvature and color relies on the accuracy of the hardware device and inevitably carries noise. (2) Although CCF strengthens the features of the majority of points, it also introduces some errors. For example, points that are not originally correspondences may be erroneously identified as inliers due to high curvature or similar color, leading to misjudgments in correspondences and thus reducing PIR and IR. However, thanks to the powerful outlier rejection ability of FCH, UPC-PCR can achieve robust registration.

**Ablation on Feature-based Compatibility Hypergraph Convolution.**    To validate the role of FCH, we report the experimental results using the FCH-based estimator, RANSAC, LGR (Qin et al., 2022) as well as deep robust estimators DGR (Choy et al., 2020) and PointDSC (Bai et al., 2021b)

Table 4: Comparison of FCH with other estimators.

| Experiment | Estimator | RR (%)↑ Color3DMatch | RR (%)↑ Color3DLoMatch |
|---|---|---|---|
| (a) | RANSAC | 97.5 | 87.4 |
| (b) | LGR | 96.9 | 87.2 |
| (c) | DGR (Choy et al., 2020) | 92.9 | 82.4 |
| (d) | PointDSC (Bai et al., 2021b) | 97.3 | 87.6 |
| (e) | FCH end-to-end | 96.6 | 90.2 |
| (f) | FCH | **98.3** | **90.4** |

Table 5: Ablation on positional embedding.

| Experiment | Positional Embedding PD | PHD | TA | CA | 3DMatch PIR (%)↑ | FMR (%)↑ | IR (%)↑ | RR (%)↑ | 3DLoMatch PIR (%)↑ | FMR (%)↑ | IR (%)↑ | RR (%)↑ |
|---|---|---|---|---|---|---|---|---|---|---|---|---|
| (a) |  | ✓ | ✓ | ✓ | 84.8 | 99.8 | 69.7 | 98.3 | 57.6 | 96.3 | 46.9 | 90.4 |
| (b) | ✓ |  | ✓ | ✓ | 88.7 | 99.9 | 72.3 | 98.3 | 61.7 | 97.2 | 49.2 | 87.7 |
| (c) |  | ✓ |  | ✓ | 83.7 | 99.8 | 68.6 | 98.0 | 55.8 | 96.2 | 45.7 | 88.6 |
| (d) |  | ✓ | ✓ |  | 83.8 | 99.7 | 69.2 | 97.5 | 55.4 | 96.3 | 46.1 | 87.6 |
| (e) | ✓ |  | ✓ |  | 87.5 | 99.3 | 72.2 | 97.3 | 59.9 | 95.0 | 48.8 | 85.7 |

in Table 4. Since the preceding networks are the same, we only need to analyze the RR metric. On the Color3DMatch dataset, since the overlap between two point clouds is high, all estimators can estimate the transformation relatively well. However, on the Color3DLoMatch dataset, they cannot perform well. Instead, FCH, which utilizes the high specificity features extracted by the preceding network and explicitly retrieves high-order compatibility of correspondences, achieves the highest RR of up to 90.4%. We also observed that compared to end-to-end training of FCH in (e), the two-stage training (f) can achieve slightly better performance. This might be due to the more significant training capability of the loss function under the two-stage training scheme.

**Ablation on Positional Embedding.** Positional embedding has a significant impact on the accuracy of point cloud registration. To investigate this, we conducted an ablation study on different combinations of embeddings, where multiple embeddings are combined with addition. The rest of the network components remained consistent, including CCF and FCH-based transformation estimator. The experimental results are presented in Table 5. The listed four types of positional embedding are Pair-wise Distance (PD) Embedding, Pair-wise Hue-Distance (PHD) Embedding, Triplet-wise Angular Embedding (TA), and our Centroid Angular (CA) Embedding, respectively. Experiment (a) corresponds to the embedding method used in UPC-PCR, which achieves the best performance in most indicators. However, compared to (b)(e), which utilizes PD instead of PHD, (a) exhibits relatively lower PIR and IR. This confirms our previous hypothesis that although using color and CA can extract more accurate superpoint features, they may lead to some incorrect correspondences. On the other hand, though using PD results in relatively higher PIR and IR, its feature ambiguity leads to lower RR, which is the most important indicator. Experiments (a)(c) validate the role of TA, (a)(b) verify the effect of PHD, and (a)(d) demonstrate the effect of CA.

## 5 CONCLUSION

In this paper, we propose UPC-PCR: Unleash the Power of Color for Point Cloud Registration. To provide more distinctive initial point features, we introduce the Curvature-Color Fusion Model to explicitly compute and fuse geometric-color information. This significantly enhances the feature distinctiveness of hierarchical points. To better encode structural information, we propose a Centroid Angular embedding, which can help detect the correspondences in the overlapping regions in low overlap registration, thus enhancing the robustness of patch matching. Finally, we design a Feature-based Compatibility Hypergraph Convolution as a bridge to connect the preceding network and the transformation estimator. It utilizes the initial features of corresponding points and aggregates them with the network. By doing this, it retrieves high-order correlations in correspondences and effectively rejects outliers. UPC-PCR achieves a significant improvement in registration performance on multiple datasets, and makes it possible to address challenging registration cases in real-world scenarios with extremely low overlap and insignificant structural characteristics.

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

APPENDIX

In this supplementary material, we first provide specific definitions of the performance metrics in the experiments (Sec. A). Then, we provide a detailed introduction to the datasets for evaluation (Sec. B), implementation details (Sec. C.1), and network architecture (Sec. C.2). Subsequently, we conduct a comprehensive evaluation of the model from various perspectives, including assessment of transformation quality (Sec. D.1), computation time and space overheads (Sec. D.2), and robustness against noise (Sec. D.3). Then, we analyze error cases encountered during algorithm execution (Sec. E.1), discuss current limitations, and propose potential improvements for future work (Sec. E.2). Finally, we present visualizations of registration results (Sec. F.1), including cases from the datasets and registration results in challenging real-world scenarios (Sec. F.2).

## A. EVALUATION METRICS

**Inlier Ratio (IR)**: The Inlier Ratio measures the proportion of point correspondences $(\mathbf{p_i}, \mathbf{q_j}) \in \hat{\mathcal{C}}$ that are within a certain residual threshold under the ground truth transformation $\overline{\mathcal{T}}_{\mathcal{P}}^{\mathcal{Q}}$. Here, $\hat{\mathcal{C}}$ denotes the estimated correspondence set between the point clouds $\mathcal{P}$ and $\mathcal{Q}$, and $\overline{\mathcal{T}}_{\mathcal{P}}^{\mathcal{Q}}$ represents the ground truth transformation from $\mathcal{P}$ to $\mathcal{Q}$. A correspondence pair is considered an inlier if the Euclidean norm of its residual is less than the threshold $\tau_1 = 10cm$. The Inlier Ratio for the point cloud pair $\mathcal{P}$ and $\mathcal{Q}$ is computed as:

$$IR(\mathcal{P}, \mathcal{Q}) = \frac{1}{|\hat{\mathcal{C}}|} \sum_{(\mathbf{p_i}, \mathbf{q_j}) \in \hat{\mathcal{C}}} \mathbb{I}\left[ \|\overline{\mathcal{T}}_{\mathcal{P}}^{\mathcal{Q}}(\mathbf{p_i}) - \mathbf{q_j}\| < \tau_1 \right], \tag{6}$$

where $\mathbb{I}[\cdot]$ is the indicator function that counts the number of correspondences with residuals less than the threshold $\tau_1$.

**Feature Matching Recall (FMR)**: The Feature Matching Recall (Deng et al., 2018a) is used to evaluate the result of feature matching by determining the fraction of point cloud pairs where the Inlier Ratio exceeds a given threshold, $\tau_2 = 5\%$. This metric reflects the probability of accurately recovering the correct transformation using the estimated correspondence set $\hat{\mathcal{C}}$, typically with the aid of a robust pose estimation algorithm such as RANSAC (Fischler & Bolles, 1981). For a dataset $\mathcal{D}$ containing $|\mathcal{D}|$ point cloud pairs, the Feature Matching Recall is defined as follows:

$$FMR(\mathcal{D}) = \frac{1}{|\mathcal{D}|} \sum_{(\mathcal{P}, \mathcal{Q}) \in \mathcal{D}} \mathbb{I}[IR(\mathcal{P}, \mathcal{Q}) > \tau_2], \tag{7}$$

where $\mathbb{I}[\cdot]$ is the indicator function that counts the number of point cloud pairs for which the Inlier Ratio exceeds the threshold $\tau_2$. This metric provides insight into the overall robustness and accuracy of the feature matching process across the entire dataset.

**Patch Inlier Ratio (PIR)**: The Patch Inlier Ratio (Qin et al., 2022) is a metric that evaluates the quality of superpoint correspondences by measuring the proportion of matches that show actual overlap when transformed using the ground-truth transformation. This metric indicates how well the estimated superpoint correspondences reflect the true geometric relationships. The Patch Inlier Ratio is calculated as follows:

$$PIR = \frac{1}{|\hat{\mathcal{C}}|} \sum_{(\hat{\mathbf{p}}_i, \hat{\mathbf{q}}_j) \in \hat{\mathcal{C}}} \mathbb{I}\left[ \min_{\tilde{\mathbf{p}} \in \tilde{\mathcal{U}}(\hat{\mathbf{p}}_i), \tilde{\mathbf{q}} \in \tilde{\mathcal{U}}(\hat{\mathbf{q}}_j)} \|\tilde{\mathbf{p}} - \tilde{\mathbf{q}}\|_2 < \tau_3 \right], \tag{8}$$

where $\tau_3 = 5cm$ is the matching radius. In this context: - $\hat{\mathcal{C}}$ denotes the estimated set of superpoint correspondences, - $\tilde{\mathcal{U}}$ is the function that up-samples a superpoint to its up-sampling points (i.e., $\tilde{\mathcal{U}}(\hat{\mathbf{p}}_i) \subset \tilde{\mathcal{P}}$), - $\mathbb{I}(\cdot)$ is the indicator function that returns 1 if the condition inside is true and 0 otherwise. This formulation ensures that the metric counts only those correspondences where at least one pair of up-sampled points from the superpoints is within the matching radius.

**Relative Translation and Rotation Errors (RTE and RRE)**: To evaluate the accuracy of the estimated transformation $\hat{\mathcal{T}}_{\mathcal{P}}^{\mathcal{Q}} \in SE(3)$, consisting of the translation vector $\hat{\mathbf{t}} \in \mathbb{R}^3$ and rotation

matrix $\hat{\mathbf{R}} \in SO(3)$, we calculate the Relative Translation Error (RTE) and Relative Rotation Error (RRE) with respect to the ground truth transformation $\overline{\mathcal{T}}_{\mathcal{P}}^{\mathcal{Q}}$ as follows:

$$RTE = \|\hat{\mathbf{t}} - \mathbf{t}\|, \tag{9}$$

$$RRE = \arccos\left(\frac{\text{trace}(\hat{\mathbf{R}}^T \mathbf{R}) - 1}{2}\right). \tag{10}$$

Here, $\mathbf{t}$ and $\mathbf{R}$ denote the ground truth translation and rotation components in $\overline{\mathcal{T}}_{\mathcal{P}}^{\mathcal{Q}}$, respectively. The RTE measures the Euclidean distance between the estimated and ground truth translation vectors, while the RRE quantifies the angular difference between the estimated and ground truth rotation matrices.

**Registration Recall (RR)**: The Registration Recall (Choi et al., 2015) is a metric used to evaluate the accuracy of point cloud registration. It measures the fraction of point cloud pairs for which the Root Mean Square Error (RMSE) is below a certain threshold, denoted as $\tau_3 = 0.2m$. For a dataset $\mathcal{D}$ containing $|\mathcal{D}|$ pairs of point clouds, the Registration Recall is defined as follows:

$$RR(\mathcal{D}) = \frac{1}{|\mathcal{D}|} \sum_{(\mathcal{P},\mathcal{Q}) \in \mathcal{D}} \mathbb{I}[RMSE(\mathcal{P},\mathcal{Q}) < \tau_3], \tag{11}$$

where $\mathbb{I}[\cdot]$ is an indicator function that counts the number of point cloud pairs with an RMSE below the threshold $\tau_3$. The RMSE for each pair $(\mathcal{P}, \mathcal{Q}) \in \mathcal{D}$ is calculated as:

$$RMSE(\mathcal{P},\mathcal{Q}) = \sqrt{\frac{1}{|\overline{\mathcal{C}}|} \sum_{(\mathbf{p_i},\mathbf{q_j}) \in \overline{\mathcal{C}}} \|\mathcal{T}_{\mathcal{P}}^{\mathcal{Q}}(\mathbf{p_i}) - \mathbf{q_j}\|^2}, \tag{12}$$

where $\overline{\mathcal{C}}$ represents the ground truth correspondences and $\mathcal{T}_{\mathcal{P}}^{\mathcal{Q}}$ denotes the estimated transformation. This metric provides an indication of the precision of the registration process across the entire dataset.

## B. DATASETS

3DMatch (Zeng et al., 2017) combines datasets from previous works such as Analysis-by-Synthesis (Valentin et al., 2016), 7Scenes (Shotton et al., 2013), SUN3D (Xiao et al., 2013), and RGB-D Scenes v.2 (Lai et al., 2014), among others. The official benchmark divides the data into 54 scenes for training and 8 scenes for testing. These scenes are captured in various indoor environments (e.g., study rooms, bedrooms, offices, living rooms) using different depth sensors (e.g., Intel RealSense, Asus Xtion Pro Live, Microsoft Kinect, etc.). We would like to acknowledge the authors of the 3DMatch dataset for making the data available under the MIT License.

Table 6: Raw data in 3DMatch (Zeng et al., 2017) and their licenses.

| Datasets | License |
| --- | --- |
| SUN3D (Xiao et al., 2013) | CC BY-NC-SA 4.0 |
| 7-Scenes (Shotton et al., 2013) | Non-commercial use only |
| RGB-D Scenes v.2 (Lai et al., 2014) | (License not stated) |
| Analysis-by-Synthesis (Valentin et al., 2016) | CC BY-NC-SA 4.0 |
| BundleFusion (Dai et al., 2017b) | CC BY-NC-SA 4.0 |
| Halber et al. (Halber & Funkhouser, 2017) | CC BY-NC-SA 4.0 |

Predator (Huang et al., 2021) and ColorPCR (Mu et al., 2024) preprocessed the original 3DMatch dataset, providing the processed source and target point cloud pairs, as well as the ground truth transformations. We utilize the preprocessed dataset Color3DMatch from ColorPCR for training and testing. Specifically, Color3DMatch is preprocessed by mapping the reshaped RGB images and depth images pixel by pixel enables the retrieval of color information for each pixel. Finally, we can convert the depth images into point clouds.

The ScanNet (Dai et al., 2017a) dataset is a large-scale RGB-D video dataset designed for 3D reconstruction, scene understanding, and semantic segmentation in indoor environments. It includes over 1,500 scans of diverse indoor scenes, featuring more than 2.5 million RGB-D images. Each scan is annotated with rich semantic and instance-level labels, covering 20 common indoor object categories. The dataset provides both raw RGB-D sequences and reconstructed 3D meshes with texture information, making it a valuable resource for advancing research in computer vision and robotics. We would like to thank the authors of the ScanNet dataset for providing the data.

## C. IMPLEMENTATION

### C.1. IMPLEMENTATION DETAILS

During the training and evaluation processes of UPC-PCR, we utilize an Intel (R) Xeon (R) CPU E5-2640 v4 and an NVIDIA GeForce RTX 3090 GPU for IO operations and computing. PyTorch (Paszke et al., 2019) serves as the primary implementation framework of UPC-PCR. For training, we employ the Adam optimizer, initialized with a learning rate of $10^{-4}$ and decayed by 5% per epoch. The batch size is configured as 1, and weight decay is set to $10^{-6}$. We employ a matching radius of 5cm, aligned with the voxel size after down-sampling, wherein point pairs within this radius are considered part of the overlapping area. We follow the data augmentation proposed in the methodology (Huang et al., 2021). During training, 128 ground-truth superpoint matches are randomly sampled, while during testing, 256 matches are sampled.

### C.2. NETWORK ARCHITECTURE

The network architecture of UPC-PCR is illustrated in Figure 5, where the three boxes from left to right represent the Feature-based Compatibility Hypergraph Convolution (FCH), the hierarchical point feature extraction backbone, and the Transformer structure for superpoint feature extraction.

**Backbone.** We propose the Curvature-Color Fusion Model (CCF) to initialize dense point features. For the input colored point cloud, we process it to extract curvature and $(h, s, v)$ values, which are then fused to obtain the initial dense point features. These dense points with initial features are subsequently input into the KPConv-FPN (Thomas et al., 2019; Lin et al., 2017) backbone network. Specifically, we follow the method described in (Thomas et al., 2019) to perform down-sampling operations to obtain hierarchical down-sampling points. The initial voxel size for the first-level down-sampling is 2.5cm, and this value is doubled for each subsequent down-sampling. Notably, We follow GeoTransformer (Qin et al., 2022) by using group normalization with 8 groups after the KPConv layers.

**Superpoint Transformer.** After obtaining the initial superpoint features through the backbone described above, we use a Transformer (Qin et al., 2022) structure to capture more precise superpoint features. To fully leverage structural information and enhance feature distinctiveness, we employ three types of positional embeddings: Pair-wise Distance Embedding, Triplet-wise Angular Embedding, and Centroid Angular Embedding, which are summed together. These embeddings are then used for self-attention on the superpoint features, followed by cross-attention between the two point clouds. This process is repeated three times, and finally, a linear layer is used to obtain the final features, which contain rich semantic information.

**FCH.** After obtaining the point correspondences, we use the FCH for feature aggregation. Each correspondence is abstracted as a vertex in a hypergraph. This vertex contains a pair of corresponding points, and we concatenate the features of these two points to obtain the initial vertex features. Next, we use the Compatibility HGNN-Conv layer to compute compatibility and perform hyperedge convolution (Bai et al., 2021a). This convolution process is repeated twice, ultimately resulting in final features that encapsulate higher-order consistency relationships.

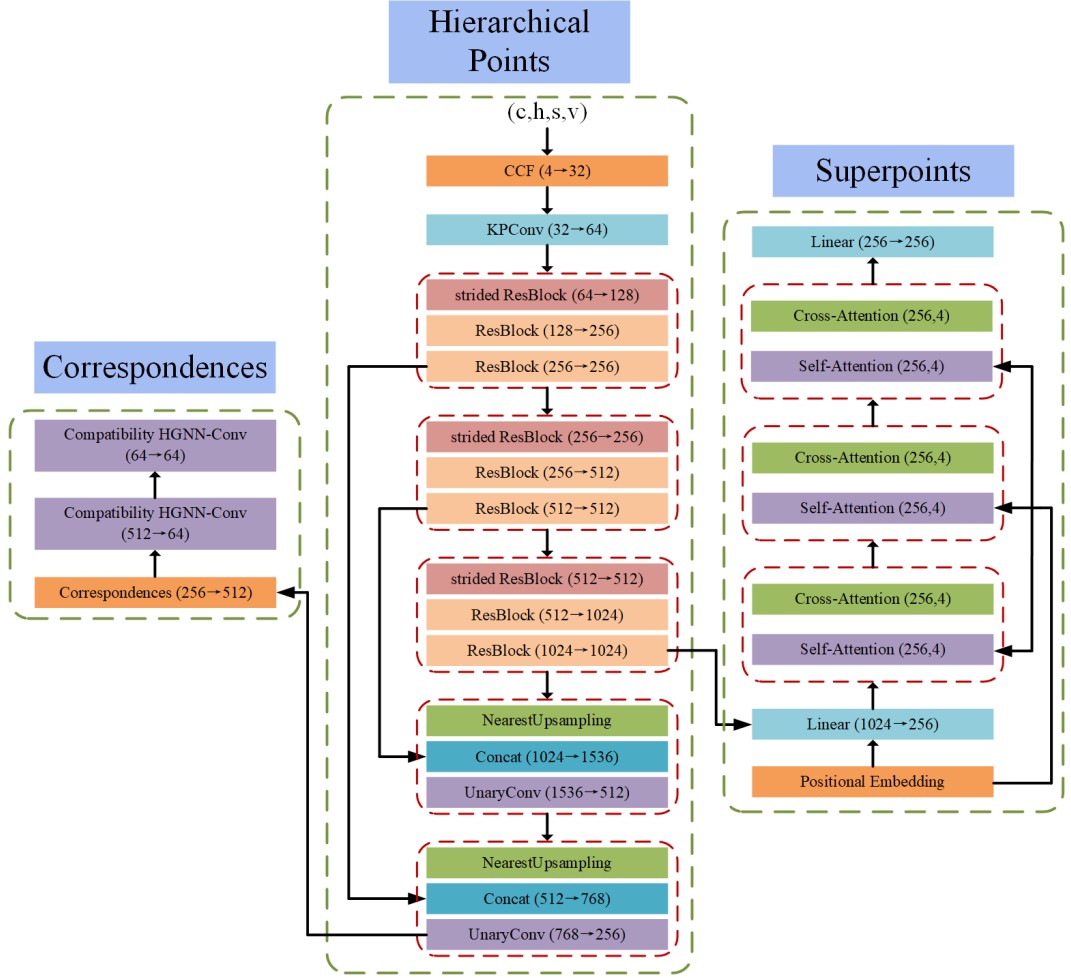

Figure 5: Network architecture of UPC-PCR.

## D. MORE EXPERIMENTS

### D.1. BASELINE COMPARISON

In Table 1 and Table 2, we compared our approach with methods such as Predator (Huang et al., 2021), CofiNet (Yu et al., 2021), and GeoTransformer Qin et al. (2022). However, these methods do not utilize color information. To facilitate a fair comparison, we concatenated the RGB color information with the geometric features (XYZ) for feature extraction. After incorporating color, we trained these methods on Color3DMatch (Zeng et al., 2017; Mu et al., 2024) and then compared them with UPC-PCR on the Color3DMatch and ScanNet (Dai et al., 2017a) datasets.

### D.2. TRANSFORMATION QUALITY

**RRE and RTE.** We compare the RRE and RTE of UPC-PCR with recent point cloud registration methods (Huang et al., 2021; Yu et al., 2021; Qin et al., 2022; Zhang et al., 2022; Yew & Lee, 2022; Yu et al., 2023b; Mu et al., 2024) on Color3DMatch/Color3DLoMatch, and the experimental results are shown in Table 7. We achieve the second-best RRE and the third-best RTE results on both Color3DMatch and Color3DLoMatch, slightly behind ColorPCR (Mu et al., 2024) and REGTR (Yew & Lee, 2022). The experiments in the main text demonstrate that UPC-PCR can achieve accurate registration in the vast majority of scenarios, even in extremely challenging low-overlap

Table 7: Relative Rotation Errors and Relative Translation Errors on 3DMatch/3DLoMatch datasets.

| Model | Color3DMatch | | Color3DLoMatch | |
|---|---|---|---|---|
| | RRE (°) | RTE (m) | RRE (°) | RTE (m) |
| Predator (Huang et al., 2021) | 2.029 | 0.064 | 3.048 | 0.093 |
| CoFiNet (Yu et al., 2021) | 2.002 | 0.064 | 3.271 | 0.090 |
| GeoTransformer (Qin et al., 2022) | 1.772 | 0.061 | 2.849 | 0.088 |
| PCR-CG (Zhang et al., 2022) | 1.993 | 0.061 | 3.002 | 0.087 |
| REGTR (Yew & Lee, 2022) | 1.567 | 0.049 | 2.827 | 0.077 |
| PEAL (Yu et al., 2023b) | 1.748 | 0.062 | 2.788 | 0.087 |
| ColorPCR (Mu et al., 2024) | **1.492** | **0.048** | **2.581** | **0.075** |
| UPC-PCR (ours) | 1.524 | 0.051 | 2.665 | 0.079 |

Table 8: Comparison of registration recall, runtime and model size.

| Model | RR (%) | | Time (s) | | | Model Size (MB) |
|---|---|---|---|---|---|---|
| | 3DMatch | 3DLoMatch | Model | Pose | Total | |
| Predator (Huang et al., 2021) | 89.0 | 59.8 | 0.052 | 3.241 | 3.293 | 56.70 |
| CofiNet (Yu et al., 2021) | 89.3 | 67.5 | 0.231 | 1.421 | 1.652 | 62.84 |
| GeoTransformer (Qin et al., 2022) | 92.0 | 75.0 | 0.126 | 2.193 | 2.319 | 37.61 |
| UPC-PCR (ours) | 98.4 | 90.4 | 0.489 | 0.094 | 0.583 | 38.13 |

cases. Furthermore, Table 7 shows that UPC-PCR can estimate tight transformations, achieving high-quality registration outcomes.

## D.3. COMPUTING OVERHEAD AND RUNTIME

We evaluate the registration time overhead and model size of UPC-PCR and compare them with recent methods including Predator (Huang et al., 2021), CofiNet (Yu et al., 2021), and GeoTransformer (Qin et al., 2022). The experimental results are shown in Table 8. In the table, Model Time refers to the time taken by the model to estimate correspondences, while Pose Time refers to the time taken to estimate the final transformation from the correspondences. All three methods compared with UPC-PCR use RANSAC-50k as the estimator. As shown, UPC-PCR significantly outperforms the baselines in registration accuracy and, being a RANSAC-free method, it is also extremely fast. Additionally, its model size is comparable to the smallest, GeoTransformer (Qin et al., 2022). These results demonstrate the robust overall performance of UPC-PCR.

## D.4. ROBUSTNESS UNDER NOISE

Our Curvature-Color Fusion Model (CCF) utilizes color information from point clouds, which are often generated through RGBD cameras. Due to limitations such as camera resolution, lighting, and scene overlap, color acquisition often introduces significant noise. To thoroughly verify UPC-PCR's robustness to color noise, we conducted noise addition experiments, as shown in Table 9.

We specifically added two types of noise to the point cloud colors (values in the range [0,1]). The first type involved adding Gaussian noise with a mean of 0 to the dense points, simulating the effects of lighting, viewing angles, and other factors on the color sampling by RGB cameras. As shown, when the noise standard deviation is small, the registration performance of UPC-PCR hardly decreases. As the noise continues to increase, there is a slight performance degradation. Even with a noise standard deviation as high as 0.5, the network can still achieve good registration, demonstrating UPC-PCR's robustness.

The second type of noise involves randomly selecting a certain proportion of dense points and assigning them completely random color values. This noise simulates the scenario where color information is incorrectly mapped to points at the wrong depth due to scene overlap. The results in the table show that when the number of selected points is small, UPC-PCR maintains stable performance. Notably, even when the proportion is as high as 30%, the model performance only experiences a

Table 9: The registration results of UPC-PCR under different types of noise.

| Noise Type | 3DMatch | | | | 3DLoMatch | | | |
|---|---|---|---|---|---|---|---|---|
| | PIR | FMR | IR | RR | PIR | FMR | IR | RR |
| Without noise | 84.8 | 99.8 | 69.7 | 98.3 | 57.6 | 96.3 | 46.9 | 90.4 |
| N(0, 0.0001) | 84.8 | 99.8 | 69.6 | 98.2 | 57.5 | 96.4 | 46.9 | 88.6 |
| N(0, 0.0005) | 84.8 | 99.8 | 69.6 | 98.2 | 57.5 | 96.4 | 46.8 | 89.7 |
| N(0, 0.001) | 84.7 | 99.8 | 69.6 | 98.3 | 57.5 | 96.3 | 46.8 | 90.0 |
| N(0, 0.005) | 84.7 | 99.8 | 69.3 | 98.1 | 57.3 | 96.4 | 46.6 | 89.2 |
| N(0, 0.01) | 84.5 | 99.7 | 68.9 | 98.1 | 57.0 | 96.5 | 46.0 | 88.9 |
| N(0, 0.05) | 82.9 | 99.7 | 65.7 | 96.8 | 54.2 | 94.4 | 42.2 | 84.5 |
| N(0, 0.1) | 80.8 | 99.5 | 62.3 | 96.1 | 50.9 | 93.2 | 38.2 | 82.4 |
| N(0, 0.5) | 60.9 | 95.4 | 39.0 | 85.1 | 28.7 | 71.9 | 18.0 | 55.0 |
| random 0.1% | 84.8 | 99.8 | 69.7 | 98.3 | 57.5 | 96.4 | 46.9 | 89.0 |
| random 0.2% | 84.8 | 99.8 | 69.6 | 98.3 | 57.4 | 96.4 | 46.8 | 88.9 |
| random 0.5% | 84.7 | 99.8 | 69.4 | 98.3 | 57.3 | 96.4 | 46.6 | 89.0 |
| random 1.0% | 84.6 | 99.8 | 69.1 | 98.1 | 57.3 | 96.5 | 46.3 | 90.1 |
| random 2.0% | 84.6 | 99.8 | 68.8 | 97.5 | 57.1 | 96.1 | 45.8 | 89.3 |
| random 5.0% | 84.0 | 99.6 | 67.4 | 97.8 | 56.4 | 96.1 | 44.5 | 88.8 |
| random 10.0% | 83.4 | 99.6 | 65.7 | 97.5 | 55.1 | 96.2 | 42.6 | 86.7 |
| random 20.0% | 82.1 | 99.5 | 62.8 | 97.2 | 52.7 | 94.5 | 39.4 | 83.2 |
| random 30.0% | 80.5 | 99.1 | 60.0 | 96.6 | 50.2 | 92.1 | 36.4 | 82.3 |

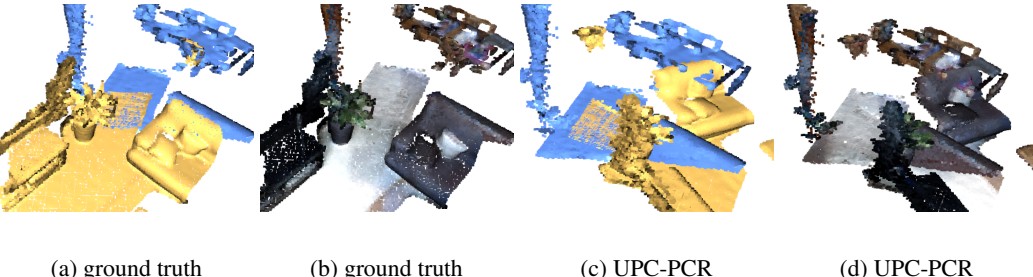

| (a) ground truth | (b) ground truth | (c) UPC-PCR | (d) UPC-PCR |

Figure 6: An example of UPC-PCR's limitation.

slight decline. These experiments strongly validate UPC-PCR's robustness to noise, indicating that it can achieve reliable point cloud registration even under conditions with significant real-world noise.

## E. SHORTCOMINGS AND PROSPECTS

### E.1. FAILURE SCENARIOS OF REGISTRATION

Although UPC-PCR significantly enhances the robustness of point cloud registration, mismatches can still occur in extremely challenging low-overlap scenarios. As shown in Figure 6, subfigures (a) and (b) represent the ground truth registration results, while (c) and (d) display the mismatches produced by UPC-PCR. In this scenario, the overlapping regions are primarily concentrated on the white floor area, with only a small part on top of the sofa. UPC-PCR successfully registers the large floor area but struggles to identify the overlapping region on the sofa top, resulting in a registration failure.

### E.2. LIMITATIONS AND PROSPECTS

UPC-PCR excels in extracting precise point-wise features and retrieving higher-order associations of correspondences in overlapping regions. However, when the overlap area is extremely small or even consists of very narrow linear regions (such as the top of the sofa in Figure 6), UPC-PCR struggles

to detect the corresponding regions. This issue limits the performance of UPC-PCR in extremely challenging scenarios.

To address the aforementioned issues, we propose the following future directions. One feasible approach is point cloud completion. For overly extreme low-overlap scenarios, conservative point cloud completion can be employed to expand the overlap regions, thereby reducing the difficulty of the point cloud registration model. Another potential method is to incorporate semantic information of the point cloud. Even if the overlap region is small, the semantic information of the adjacent regions should be similar. Therefore, explicitly using semantic information can guide the transformation estimation process.

## F. VISUALIZATION

### F.1. QUALITATIVE RESULTS

We provide qualitative registration results on the RGBD datasets 3DMatch (Figure 7) and ScanNet (Figure 8). In both figures, (a) represents the poses of the two input point clouds; (b) shows the ground truth registration results, with both color and colorless visualizations; (c) shows the registration results of UPC-PCR.

### F.2. REAL WORLD SCENARIOS

We also provide UPC-PCR's qualitative registration results in real-world settings, including both indoor and outdoor scenes, as shown in Figure 9. In the figure, (a) represents the target point cloud, (b) represents the source point cloud, (c) shows the initial poses of the two input point clouds, and (d) displays UPC-PCR's registration results.

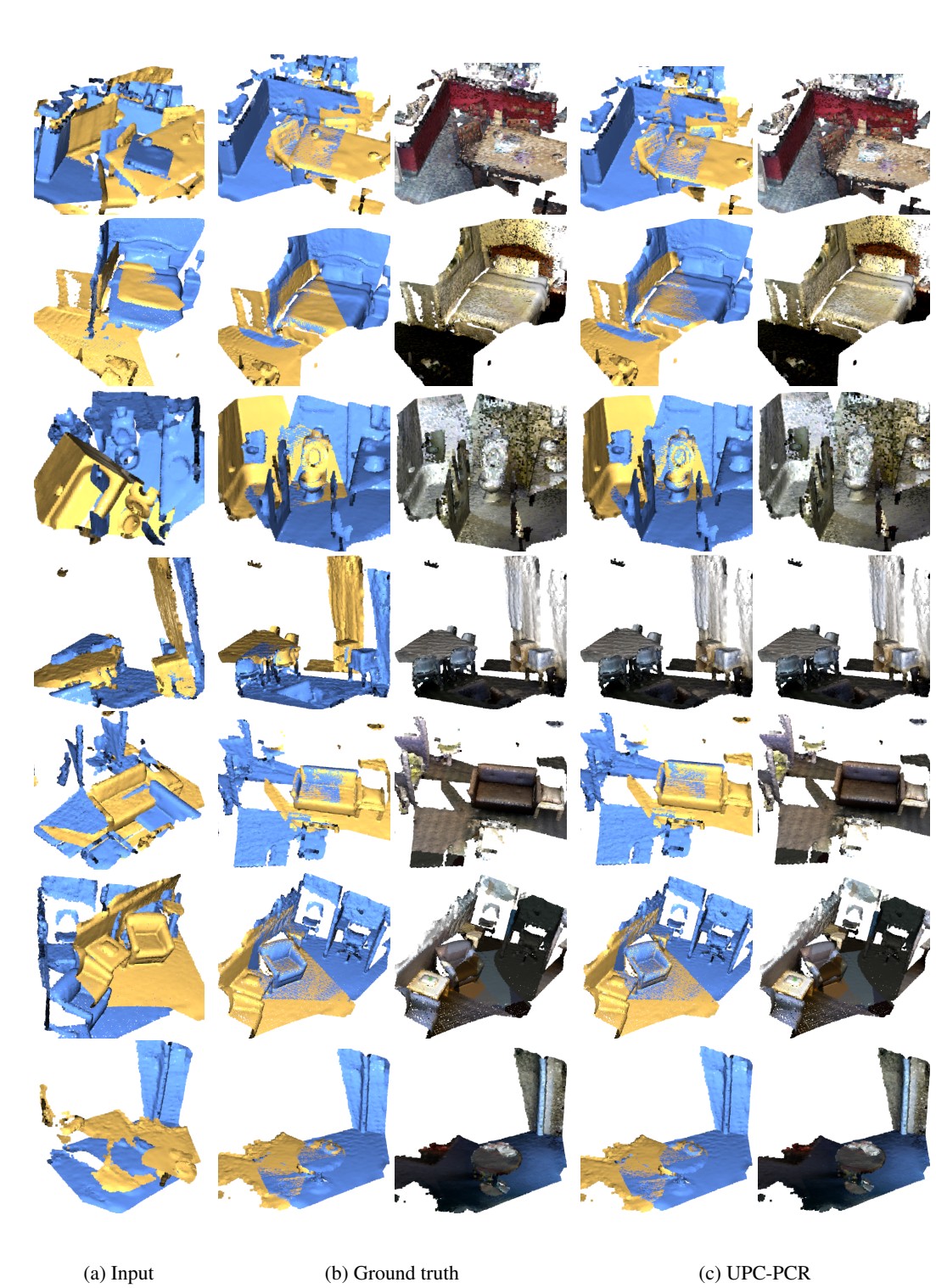

(a) Input            (b) Ground truth            (c) UPC-PCR

Figure 7: Qualitative registration results on Color3DMatch.

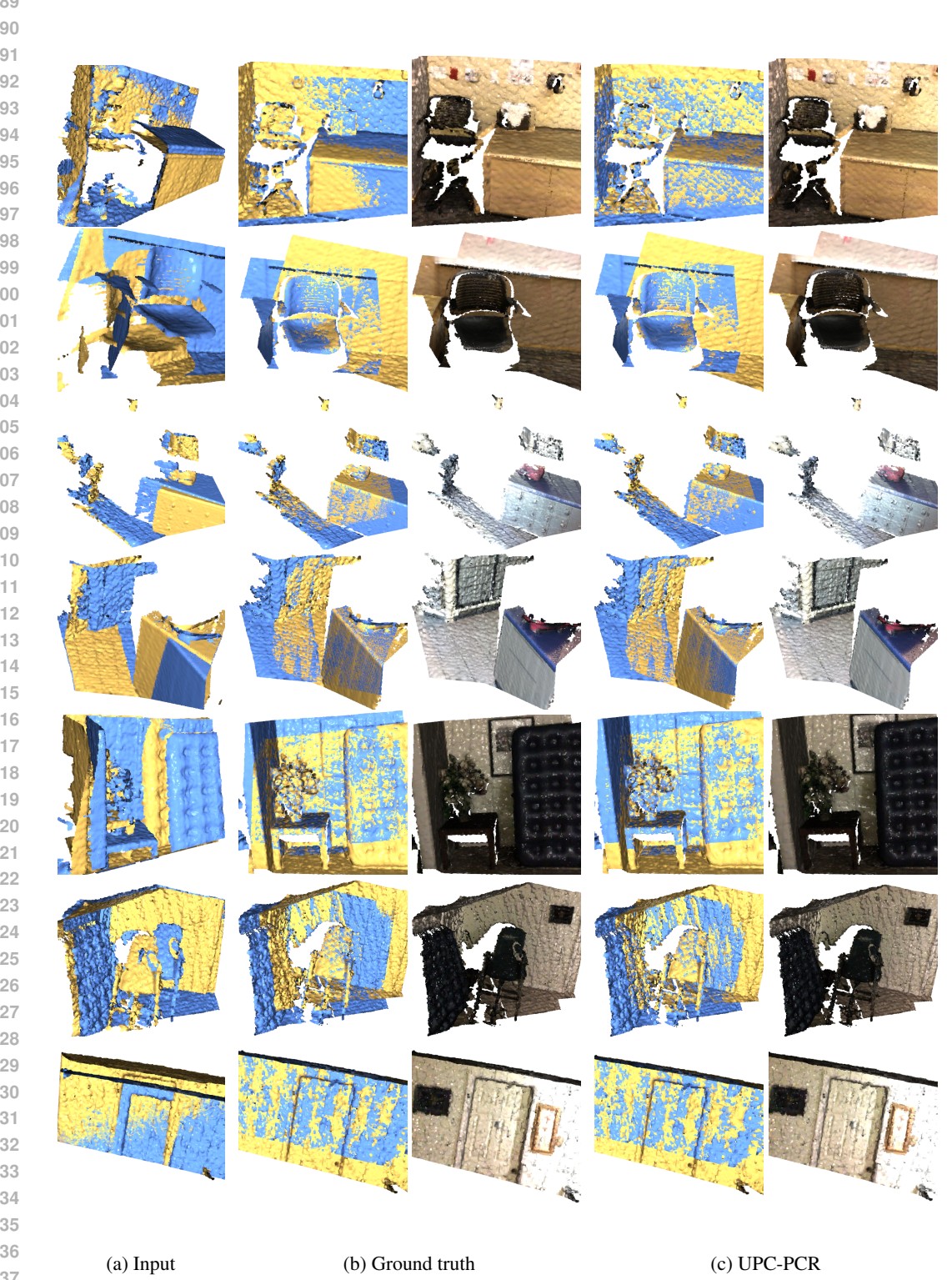

(a) Input          (b) Ground truth          (c) UPC-PCR

Figure 8: Qualitative registration results on ScanNet.

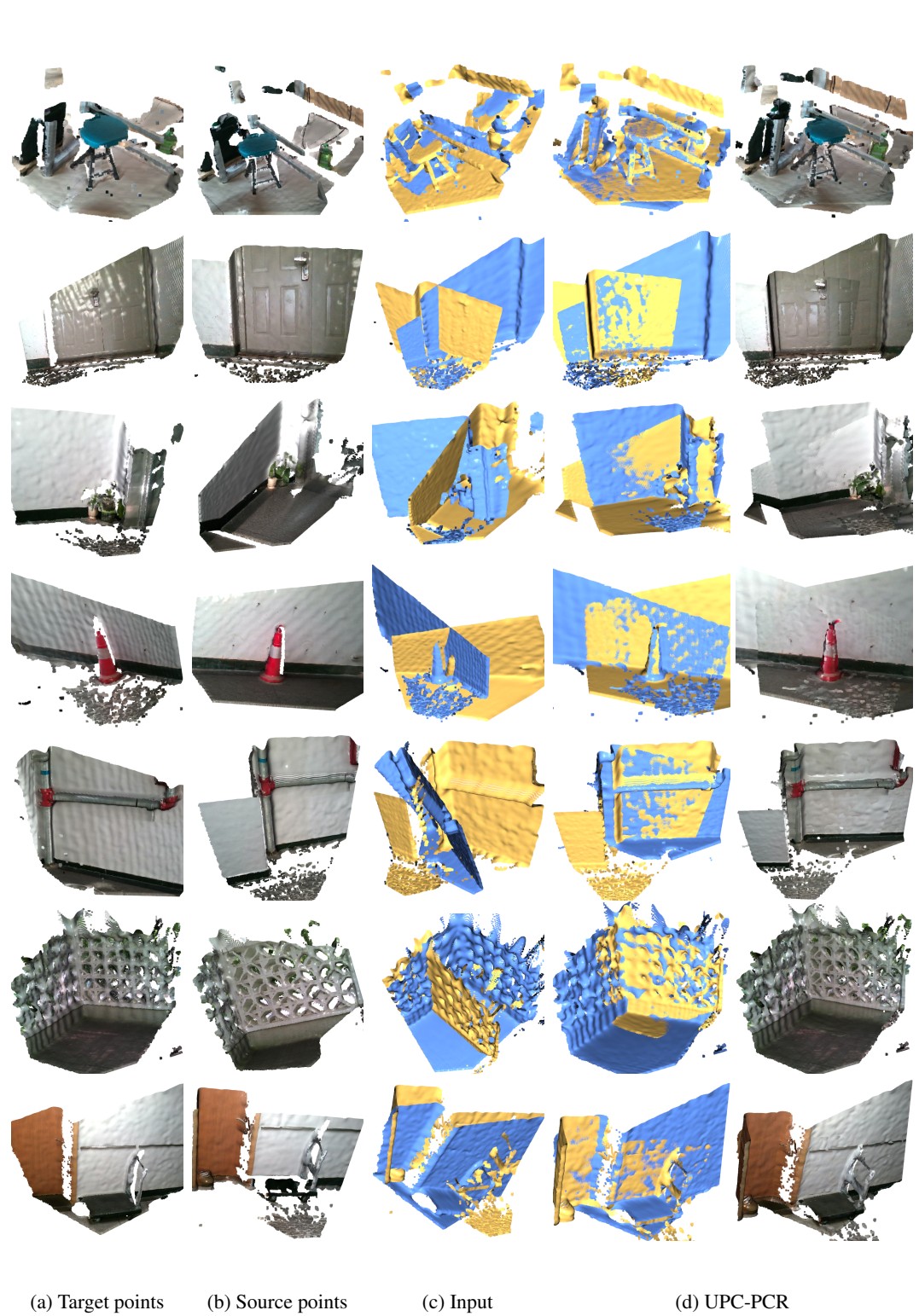

(a) Target points      (b) Source points      (c) Input            (d) UPC-PCR

Figure 9: Qualitative registration results in real-world indoor and outdoor scenes.

