# OpenReview forum: "Unleash The Power of Color for Point Cloud Registration"
_ICLR.cc/2025/Conference — ICLR 2025 Conference Withdrawn Submission_

### Official Review · Reviewer_zegw · 2024-10-22

**Soundness:** 3
**Presentation:** 2
**Contribution:** 2
**Rating:** 5
**Confidence:** 4

**Summary:**

This article studies the color involved PCR. The authors propose several tactics to better recruite colors, including the Curvature-Color Fusion Module (CCF) for feature initialization, Centroid Angular (CA) included positional embeddings for structural consistency and the Feature-based Compatibility Hypergraph Convolution (FCH) for outlier removal. Experimental results show the efficacy of the presented method. In general, the paper is easy to follow and presentation is fine.

**Strengths:**

The article introduces a pipeline to do colored point coloud registration with several novel ideas. CCF considers the fusion of color and curvature, which technically adopts color for better feature extraction. CA was believed to help counter the small overlaps. FCH can learn higher-order compatibilities  through feature aggregation for outlier rejection.

**Weaknesses:**

The biggest concern for me is that the color is effectively explored by the authors. They claim that color can be powerful in strengthing the capabilities of registration methods. However, what they adopts seems just a simple fusion in CCF. I don't think the power of color is effectively explored by the authors.

The minor concerns are on the three modules introduced, CCF, CA and FCH. They are more or less incremental or not very significant. FCH seems a bit more creative than other two in my view, but it is basically a compatibility integrated graph convolution.

**Questions:**

1. The authors aim at exploring the color for better performances, but actually the proposed method only takes the color in the very first fusion step. In addition, the usage is also quite simple by fusion. I don't see they have fully explored the power of color. In other words, the power of the proposed method still relies on the 3D geometrical points.

2. CA is said to tackle the small overlapping problem because it may change little for small overlaps. However, I don't understand how it can serve as positional guidance here. It is perhaps yet another information on the points. In addition, the ablation on CA seems confusing (Tab. 5). It may not so good as expected as IR shows.

3. FCH is also a bit difficult to understand. Perhaps a figure introduction can solve all my coming confusions. It seems that the compatibilities are used to create the hypergraph for  further feature aggregation. However, what's the principle behind Eq. 3 and why can the correlations be measured by Eq. 5? I can only barely understand them.

4. Overall, the proposed tactics, CCF, CA and FCH, are not very significant. I believe they can be useful for the color PCR  but they are not very significant but just incremental contributions. None of them is deeply explored for maximal performances. This feeling can also be observed from the results which are not very stable and consistent.

---

### Official Review · Reviewer_Uh7q · 2024-10-28

**Soundness:** 3
**Presentation:** 3
**Contribution:** 3
**Rating:** 5
**Confidence:** 3

**Summary:**

This paper proposes a point cloud registration pipeline. The main contribution of this paper contains a module that combines color information with geometric information; a new positional embedding method; and a hypergraph convolution method to learn higher-order compatibility among correspondences for outlier rejection. The proposed method is compared with different baselines including color or without color point cloud input, and RGB-D image input, and achieves good performances compared with existing baseline methods.

**Strengths:**

The direct combination color and geometric information has merit and seems novel.
The proposed modules seem effective based on the ablation studies.

**Weaknesses:**

The figures are of lower quality and should be upgraded to higher resolution ones.

Novelty: there are point cloud registration methods based on curvature features. The only difference between the proposed FCH module and the one used in ColorPCR seems to be the compatibility function. Then only the CA embedding is the novel module. I also suggest the authors add the comparison of FCH with ColorPCR in ablation studies.

The title is UNLEASH THE POWER OF COLOR FOR POINT CLOUD REGISTRATION, however, the only operation for color information is the concatenation with geometric information. In ablation studies, the authors also mentioned that FCH helps reject the errors caused by color. And the results of the proposed method do not totally overperform ColorPCR, which the authors described as ‘imposes certain limitations on the utilization of color power’. Thus the color does not seem to be very important.

There are a few typos in Table 1 and Table 2.

**Questions:**

If one of the functions of FCH module is to help reject the misjudgments due to high curvature or similar color, should the design of FCH also combines color or curvature?

---

### Official Review · Reviewer_GSik · 2024-11-01

**Soundness:** 2
**Presentation:** 2
**Contribution:** 2
**Rating:** 3
**Confidence:** 5

**Summary:**

This paper introduces color information for the aiding of point cloud registration, uses curvature-color fusion and centroid angular embedding to initialize the correspondences, and develops a feature-based compatibility hypergraph convolution to refine correspondences. Experiments on benchmark datasets including Color3DMatch/Color3DLoMatch validate the efficacy of each component of the method.

**Strengths:**

1. The paper is well-written and easy to follow.

2. The motivation is clearly clarified.

**Weaknesses:**

1. From a holistic viewpoint, the current study represents more of an incremental step compared to previous methods, notably relying heavily on GeoTransformer [1] and ColorPCR [2]. Consequently, I find the technical contributions somewhat constrained. For example, while the authors introduce curvature to the prior color feature [2], curvature has also been extensively utilized, as seen in [3]. The centroid angular embedding utilized in this study appears to be a mere incremental extension of pair-wise distance embedding and triplet-wise angular embedding, retaining a similar structural framework.

2. On line 363 of Page 7, the author mentions, "For IR, because of the aggressive introduction of color, which may disrupt the spatial structure of the point cloud." I am curious as to why color is perceived to disrupt geometric structures.

3. In Table 2, why does the translation accuracy not match up to ColorPCR overall? Could this be attributed to the inclusion of color information?

4. In Table 3, I recommend that the authors include all metrics instead of solely RR, as this could offer a more comprehensive evaluation of the current method.

5. Another concern I have is regarding benchmark testing. Given that the current registration recall and matching recall values are nearly 100%, have the authors tested real-world scanned point clouds, such as data captured from LiDAR or autonomous driving? This practical testing may hold more significance than mere quantitative enhancements.

6. Typos:

(1). Line 064 Page 2, “Unleash the power”->unleash

(2). Line 171 Page 4, “We”->we

(3). Line 295 Page 6, “The”->the

(4). Line 313 Page 6, “while in COlor3DLoMatch”->Color3DLoMatch





References:

[1] Zheng Qin, Hao Yu, Changjian Wang, Yulan Guo, Yuxing Peng, and Kai Xu. Geometric transformer for fast and robust point cloud registration. In IEEE Conf. Comput. Vis. Pattern Recog., pp. 11143–11152, 2022.

[2] Juncheng Mu, Lin Bie, Shaoyi Du, and Yue Gao. Colorpcr: Color point cloud registration with
multi-stage geometric-color fusion. In IEEE Conf. Comput. Vis. Pattern Recog., pp. 21061–21070,
2024.

[3] Radu Bogdan Rusu and Steve Cousins. 3d is here: Point cloud library (pcl). In IEEE international conference on robotics and automation, pp. 1–4, 2011.

**Questions:**

Please see the weakness part.

---

### Official Review · Reviewer_L9Jk · 2024-11-04

**Soundness:** 1
**Presentation:** 1
**Contribution:** 1
**Rating:** 3
**Confidence:** 5

**Summary:**

This paper proposes a point cloud registration pipeline with color information. It proposes a Curvature-Color Fusion Module to initialize point cloud features, then proposes a Centroid Angular embedding for superpoint structure encoding. Finally, it proposes Feature-based
Compatibility Hypergraph Convolution to learn higher-order compatibility of correspondences  for filtering out outliers. The experimental results demonstrate that the proposed method could perform well on Color3DMatch and Color3DLoMatch datasets.

**Strengths:**

1. High performance is achieved on Color3DMatch and Color3DLoMatch datasets.

**Weaknesses:**

1. This paper lacks novelty. For example, the proposed Curvature-Color Fusion Module is simply a fusion of color information and point cloud structural information, which is quite similar to existing approaches in the literature[1],[2],[3]. The Centroid Angular embedding, however, is confusing to me. Since it is one of the core modules, I believe the authors should clearly explain how it functions. As for Feature-based Compatibility Hypergraph Convolution, it is essentially the same as what is described in Hunter[4]. Therefore, I think that this paper is not sufficiently innovative enough to be accepted by ICLR.
2. This work appears to be a systematic project that combines various existing works. Its contribution to advancing the community is incremental, more akin to small improvements or "tricks" rather than groundbreaking innovations.
3. When the inlier ratio is significantly lower, it’s puzzling that the registration recall rate can still be higher. The authors should provide an analysis to explain this.

[1] Zhang Y, Yu J, Huang X, et al. Pcr-cg: Point cloud registration via deep explicit color and geometry[C]//European Conference on Computer Vision. Cham: Springer Nature Switzerland, 2022: 443-459.
[2] Yu J, Ren L, Zhang Y, et al. PEAL: Prior-Embedded Explicit Attention Learning for Low-Overlap Point Cloud Registration[C]//Proceedings of the IEEE/CVF Conference on Computer Vision and Pattern Recognition. 2023: 17702-17711.
[3]Mu J, Bie L, Du S, et al. ColorPCR: Color Point Cloud Registration with Multi-Stage Geometric-Color Fusion[C]//Proceedings of the IEEE/CVF Conference on Computer Vision and Pattern Recognition. 2024: 21061-21070.
[4] Yao R, Du S, Cui W, et al. Hunter: Exploring high-order consistency for point cloud registration with severe outliers[J]. IEEE Transactions on Pattern Analysis and Machine Intelligence, 2023.

**Questions:**

See Weaknesses.

**Details Of Ethics Concerns:**

None.

---

### Note · Authors · 2024-11-15

I have read and agree with the venue's withdrawal policy on behalf of myself and my co-authors.